# Preparation of Electrospun Small Intestinal Submucosa/Poly(caprolactone-*co*-Lactide-*co*-glycolide) Nanofiber Sheet as a Potential Drug Carrier

**DOI:** 10.3390/pharmaceutics13020253

**Published:** 2021-02-11

**Authors:** Nguyen Thi Thu Thao, Surha Lee, Gi Ru Shin, Youngji Kang, Sangdun Choi, Moon Suk Kim

**Affiliations:** Department of Molecular Science and Technology, Ajou University, Suwon 16499, Korea; thao0705utb@ajou.ac.kr (N.T.T.T.); lsh5969@ajou.ac.kr (S.L.); mirage1008@ajou.ac.kr (G.R.S.); nanobiokkang@ajou.ac.kr (Y.K.); sangdunchoi@ajou.ac.kr (S.C.)

**Keywords:** electrospinning, nanofiber sheets, small intestine submucosa, drug, sustained delivery

## Abstract

In this work, we chose small intestine submucosa (SIS) as a drug carrier because SIS possesses good biocompatibility, non-immunogenic property and bio-resorbability, and performed electrospinning for preparation of nanofiber sheets (NS). For the preparation of drug-loaded electrospun SIS nanofiber sheets as a drug carrier, we used poly(ε-caprolactone-*ran*-l-lactide) (PCLA) copolymers to improve the electrospinning performance of SIS. The electrospinning of SIS and PCLA provided the electrospun SIS/PCLA (S/P)-nanofiber sheet (S/P-NS) with adjustable thickness and areas. The electrospun S/P-NS showed different porosities, pore sizes, diameters and tensile strengths depending on the ratios between SIS and PCLA. The electrospun S/P-NS was used as a drug carrier of the dexamethasone (Dex) and silver sulfadiazine (AgS) drug related to anti-inflammation. Dex-loaded S/P-NS and AgS-loaded S/P-NS was successfully fabricated by the electrospinning. In the in vitro and in vivo release, we successfully confirmed the possibility for the sustained release of Dex and AgS from the Dex-S/P-NS and AgS-S/P-NS for three weeks. In addition, the sustained Dex and AgS release suppressed the macrophage infiltration. Collectively, we achieved feasible development of SIS nanofiber sheets for a sustained Dex and AgS delivery system.

## 1. Introduction

Small intestine submucosa (SIS) derived from the submucosal intestinal layer is an acellular, collagen-based extracellular matrix (ECM) with over 90% collagen content [1]. For several decades, SIS has been commonly harvested from pigs and is one of the most studied biomaterials in many biomedical applications [2,3,4]. SIS is an ideal biomaterial because of its good biocompatibility, non-immunogenic property, and bio-resorbability. Based on these key characteristics, some biomedical products (see Cook Biotech, Vetrix BioSIS, etc.) using SIS have been developed and commercialized [5]. Additionally, SIS has been approved by the Food and Drug Administration (FDA) for several clinical products [5,6].

Recently, our research group and others have investigated SIS sheets, sponges, and hydrogels for biomedical applications [7,8,9]. Although SIS is an excellent candidate for biomedical applications, the utilization of SIS as a drug carrier is limited.

Nanofiber sheets (NS) can be used in various biomedical applications such as drug carriers, wound healing, scaffolding for tissue engineering, etc. [10,11]. Over the past few years, the development of NS for biomedical applications has grown remarkably. Among several fabrication techniques for NS, electrospinning is one of the simplest and cost-effective methods [12,13,14,15,16,17]. Successful NS electrospinning depends on the appropriate selection of solvents and polymers.

Various synthetic and natural materials can be used to fabricate electrospun NS [18,19]. Among the several options for materials, it was expected that electrospinning SIS would generate electrospun SIS nanofiber sheets (SIS-NS). However, electrospinning of SIS demonstrates the limitations of common electrospinning solvents (e.g., 1,1,1,3,3,3-hexafluoro-2-propanol) for controlling the physical properties of electrospun SIS-NS, etc.

On the other hand, synthetic polyesters such as poly(lactic acid) (PLA), poly(glycolide) (PGA), poly(d,l-lactic-*co*-glycolic acid) (PLGA), and poly(ε-caprolactone) are widely used for electrospinning because they make electrospinning relatively easy, and they can be used to easily adjust the properties of the electrospun polyesters [20,21,22].

Therefore, we hypothesized that an electrospinning solvent that is commonly used for synthetic polyesters could be applied to the electrospinning of SIS and synthetic polyesters to improve the physical properties of electrospun SIS-NS. Among several synthetic polyesters, poly(ε-caprolactone-*ran*-l-lactide) (PCLA) was selected due to several potential advantages such as the ease of preparation and controllable biodegradability and biocompatibility in vivo [23,24].

Thus, in the first part of this study, electrospinning was performed with the addition of various concentrations of PCLA into SIS in order to improve the electrospinning performance of SIS and to modulate the physical properties of the electrospun SIS/PCLA (S/P) nanofiber sheet (S/P-NS).

Dexamethasone 21-phosphate disodium salt (Dex), a type of corticosteroid medication, is used to treat several skin diseases, inflammatory and auto-immune disorders, and rheumatic disorders as well as for adjunctive therapy for short-term administration [25]. Silver sulfadiazine (AgS) is a topical antibiotic used on partial-thickness and full-thickness burns to prevent infection [26]. The structural differences between Dex and AgS cause different solubility in water. Dex is easy to dissolve in water and therefore suitable for rapid administration and repeated administration, as required [27]. AgS is less water-soluble, and thus is only suitable for very limited administration [28].

To expand the utilization of Dex or AgS, a novel delivery system is needed to enable the sustained administration of Dex or AgS. In light of this, the second aim of this study was to prepare Dex-loaded SIS/PCLA (S/P) nanofiber sheets (Dex-S/P-NS) and AgS-loaded SIS/PCLA (S/P) nanofiber sheets (AgS-S/P-NS) for a sustained Dex and AgS delivery system.

Dex-S/P-NS and AgS-S/P-NS would provide a means of maintaining concentrations of Dex and AgS within their implanted positions for a defined period under physiological conditions, as well as evaluating the biocompatibility of the Dex-S/P-NS and AgS-S/P-NS over a defined period (Figure 1). More information on these research topics will enable the development of suitable nanofiber sheets for a sustained Dex and AgS delivery system.

## 2. Materials and Methods

### 2.1. Materials

The initiator methoxy poly(ethyleneglycol) (MPEG, average molecular weight Mn = 750 g/mol) and the catalyst Sn(Oct)_2_ were purchased from Sigma Aldrich (St. Louis, MO, USA) and used without further purification. Lactide (LA; Boehringer Ingelheim, Blanquefort, France) was recrystallized twice in ethyl acetate, and ε-caprolactone (CL; Aldrich, Milwaukee, WI, USA) was distilled over CaH_2_ under reduced pressure. 1,1,1,3,3,3-Hexafluoro-2-propanol (HFIP, >99%) was obtained from Fluka (Sigma-Aldrich, MO, USA) as an electrospinning solvent. Collagenase from Clostridium histolyticum (C7657, >1200 CDU/mg) was purchased from Sigma Aldrich. Dexamethasone 21-phosphate disodium salt (Dex) and silver-sulfadiazine (AgS) were purchased from Sigma-Aldrich (St Louis, MO, USA). Other materials were used without further purification.

### 2.2. Preparation of Small Intestinal Submucosa Powder

Native SIS sheets were prepared using previously reported methods [3]. The native sheets were then crushed into powder by a freezer mill (6700, SPEX; Metuchen, NJ, USA) in liquid nitrogen. The SIS powder was treated in 0.1% pepsin and 3% acetic acid for 48 h and then neutralized by 1 M NaOH solution. Next, the neutralized SIS suspension was freeze-dried, and this superfine SIS powder was used in subsequent experiments.

### 2.3. Synthesis of PCLA Copolymer

PCLA copolymers were prepared using previously reported methods [23]. The structure and molecular weight of PCLA were determined using ^1^H NMR (Varian Mercury Plus 400, Varian, Palo Alto, CA, USA) by comparing the total methylene protons in PCL and the methyl proton signals of PLA with the total methylene protons in MPEG as a standard of 750 g/mol.

### 2.4. Preparation of Electrospun SIS/PCLA (S/P) Nanofiber Sheets (S/P-NS)

SIS and PCLA (15 wt%) were solubilized using 1,1,1,3,3,3-hexafluoro-2-propanol (HFIP). Three ratios of SIS and PCLA solutions (S:P ratio = 5:1, 3:1, and 1:1) were prepared for electrospinning. An SIS solution, PCLA solution, and three ratios of SIS and PCLA solution were individually loaded in a syringe with a 20-G needle tip. The electrospinning of homemade ES-1 Robot (DaeLim Starlet, Siheung, Korea) was performed at a voltage of 25 kV, 40% humidity, at room temperature, and with 10 cm distance between the collector and the 20-G needle tip. The flow rate of each solution was maintained at 10 mL/min using a syringe pump. The steel cylinder of the collector was wrapped in aluminum foil. The velocity of the collector was maintained at a constant speed of 300 rpm. Each electrospun S/P-NS on the surface of the collector was dried in a vacuum chamber to completely remove any remaining solvent and then stored in a vacuum desiccator. To confirm the presence of SIS and PCLA inside electrospun S/P-NS, S/P-NS solubilized in CH_2_Cl_2_. PCLA was confirmed in the CH_2_Cl_2_-soluble fraction by ^1^H NMR (Appendix A). The ratio of the insoluble and soluble portion in CH_2_Cl_2_ was almost 5 versus 1.

To prepare the electrospun Dex- or AgS-loaded SIS/PCLA (S/P) nanofiber sheets (Dex-S/P-NS and AgS-S/P-NS, respectively), Dex or AgS (3 wt%, relative to the total weight of SIS and PCLA (for the S:P ratio of 5:1)) was dissolved in SIS/PCLA (S:P = 5:1) solution. Then, the electrospinning was conducted to produce drug-loaded S/P-NS under the same conditions as the S/P-NS described in the previous paragraph. The Dex and AgS inside electrospun Dex-S/P-NS and AgS-S/P-NS were confirmed in the ^1^H NMR and DSC (Appendix A).

### 2.5. Characterization of Electrospun S/P-NS

The image of the electrospun S/P-NS was observed using scanning electron microscopy (SEM, S2250N, Hitachi, Tokyo, Japan). Each electrospun S/P-NS was sputter-coated with a gold conductive layer using a plasma sputtering apparatus (Emitech, K575, Kent, UK). The fiber diameters of the S/P-NS were measured using ImageJ software from images obtained with SEM. The average fiber diameters of the S/P-NS were calculated from measurements of 100 random fibers. A statistical analysis was performed using SPSS 16.0 with *p* < 0.05.

The static water contact angles of each electrospun S/P-NS were measured via the sessile drop method at room temperature with an optical bench-type contact angle goniometer (KRUSS DSA10-MK). One droplet of purified water (40 µL) was deposited onto each electrospun S/P-NS with a micro-syringe attached to the goniometer and then analyzed using the ImageJ software. The contact angle was measured five times and the average value was calculated.

The porosity of each electrospun S/P-NS was measured using a porosimeter (IV 9500 V 1.03 Auto Pore, Instruments Co., Norcross, GA, USA). Each electrospun S/P-NS was prepared with a weight of 100 mg. Each specimen was immersed in Hg, and then accelerating pressure from 0.2 to 45 psia was applied. All of the electrospun S/P-NS were simultaneously measured in separated ports.

The tensile strength of each electrospun S/P-NS was measured using a Universal Testing Machine (H5KT, Tinius-Olsen, Horsham, PA, USA). A dumbbell shape (with a thickness of 0.5 mm, a middle section of 5 × 5 mm^2^, and a total area of 15 × 30 mm^2^) was prepared and gripped at each vertical end of the NS. It was then pulled vertically at a crosshead speed of 10 mm/min with a 100 N load cell until a break-off was recorded. All of the experiments were performed with three specimens, and the results were statistically analyzed using SPSS 16.0 with *p* < 0.05.

### 2.6. In Vitro Degradation Test

The electrospun S/P-NS were cut into round samples using a 12-mm biopsy punch. Each specimen was weighed and then placed into a 20-mL vial. Each specimen in a vial was hydrated for 20 min with 0.5 mL PBS. Then, 5 mL collagenase (1200 CDU/mg) dissolved in a solution of PBS buffer (pH 7.4) and 5 mM CaCl_2_ were added into each vial. The vials were shaken in a humidified incubator at 37 °C and 5% CO_2_ on a shaker at 100 rpm. At each time point, each vial was carefully washed with distilled water (DW), pre-frozen overnight in a deep freezer, and lyophilized until it reached a constant weight. The degradation ratio of electrospun S/P-NS was calculated by the following equation:degradation ratio (%) = W_1_/W_0_ × 100 (%)(1)
where W_0_ is the initial weight of each specimen in the vial, and W_1_ is the weight of each specimen completely dried at each time point.

### 2.7. In Vitro Drug Release from Drug-Loaded S/P-NP

The Dex-S/P-NS and AgS-S/P-NS (drug content of 3 wt%) were immersed in 4 mL PBS (pH 7.4) in a 5-mL vial and incubated at 37 °C with shaking at 100 rpm for 21 days. At each time point, 0.5 mL of solution in the vials was removed and replaced with fresh PBS (incubated at 37 °C) of the same volume. Three independent release experiments were performed for each Dex-S/P-NS and AgS-S/P-NS (Appendix A).

The amount of Dex released was analyzed using high-performance liquid chromatography (HPLC; Agilent 1200 series, Waldbronn, Germany) at a detection wavelength of 220 nm and a C18 column (250 mm × 4.6 mm i.d., 5 µm particle size). The mobile phase was prepared as a mixture of DW and acetonitrile at a ratio of 60:40 (*v*/*v*) and eluted at a flow rate of 1.0 mL/min. Standard calibration curves were prepared using known concentrations of Dex, and these were used to quantify the Dex released from each Dex-S/P-NS.

To determine the amount of AgS released, UV/Vis photo-spectroscopy of AgS was conducted using whole spectrum scans to determine the maximum absorbance at 297 nm. Standard calibration curves were prepared using known concentrations of pure AgS to quantify the AgS released from each AgS-S/P-NS. The in vitro drug release experiment was performed in quadruplicate. The cumulative amount of released Dex and AgS was then calculated. The percentages of Dex or AgS released from Dex-S/P-NS and AgS-S/P-NS over time were plotted.

### 2.8. In Vivo Implantation

The protocols of this study were approved by the Institutional Animal Experiment Committee at Ajou University School of Medicine (No. 2018-0023). Six-week-old Sprague–Dawley (SD) rats (270–350 g) were was anesthetized with zoletil and rompun (1:1 ratio, 2 mL/kg). S/P-NS, Dex-S/P-NS, and AgS-S/P-NS were prepared in a round, 12-mm diameter shape. The S/P-NS, Dex-S/P-NS, and AgS-S/P-NS were implanted subcutaneously under the dorsal skin of the rats and allowed to develop in vivo over 3 weeks. At each post-implantation sampling point, the rats were sacrificed, and the Dex-S/P-NS and AgS-S/P-NS were removed individually from the subcutaneous dorsum for in vivo drug release and histological analysis.

### 2.9. Determination of In Vivo Drug Release

The removed S/P-NS, Dex-S/P-NS, and AgS-S/P-NS at each time point were placed in individual test tubes and then immediately lyophilized. The lyophilized S/P-NS, Dex-S/P-NS, and AgS-S/P-NS were dissolved in HFIP and then filtered to remove the insoluble part. Next, the soluble specimen was evaporated and dried to remove the HFIP. The amount of released Dex or AgS at each time point was determined using HPLC and UV/Vis, respectively, as described in Section 2.7. The in vivo drug release experiment was performed using three animals. The amounts of Dex or AgS released in vivo were determined based on the difference between the initial Dex or AgS amount in the Dex-S/P-NS or AgS-S/P-NS and the remaining Dex in the Dex-S/P-NS or AgS in the AgS-S/P-NS removed from the rats at each time point.

### 2.10. Histological Analysis

At 1, 2, and 3 weeks, the implanted S/P-NS, Dex-S/P-NS, and AgS-S/P-NS were removed, fixed with 10% formaldehyde, and embedded in paraffin. The embedded S/P-NS, Dex-S/P-NS, and AgS-S/P-NS were sectioned into 4-µm slices along the longitudinal axis of the NS.

The paraffin on the S/P-NS, Dex-S/P-NS, and AgS-S/P-NS slides was removed through incubation for 2 h at 80 °C, and then the samples were hydrated sequentially using 100%, 95%, 80%, and 70% ethyl alcohol. The S/P-NS, Dex-S/P-NS, and AgS-S/P-NS slides were rinsed in running tap water, stained with hematoxylin (Muto Pure Chemicals, Tokyo, Japan) for 3 min, and then rinsed with DW. Next, the hematoxylin-stained slides were stained with eosin for 2 min, rinsed with DW, and hydrated using 95% and 100% ethyl alcohol and xylene for 2 min. The slides were then fixed with mounting medium (Muto Pure Chemicals, Tokyo, Japan).

For ED-1 (mouse anti-rat CD68; Serotec; Oxford, UK) staining as a marker for macrophages, each deparaffinized specimen was blocked with 10% BSA (Roche, Penzberg, Germany) in PBS for 1 h at 37 °C. The specimens were incubated at 48 °C for 18 h with the ED-1 antibody (1:1000) and then treated with secondary antibody (rat anti-mouse Alexa Fluor594; Invitrogen, Carlsbad, CA) for 1 h at room temperature. The nuclei were stained with 6-diamino-2-phenylindoadihydrochloride (DAPI; Sigma) and mounted with fluorescent mounting solution (DaKo; Glostrup, Denmark). Immunofluorescence images were produced using an Olympus IX81 fluorescent microscope (Olympus; Tokyo, Japan) equipped with Meta Image Series software (MetaMorph, Molecular Devices Corporation; Downingtown, PA). The overall DAPI-labelled cell distribution within the S/P-NS, Dex-S/P-NS, and AgS-S/P-NS, and the percentage of ED-1-positive cells were calculated using ImageJ software at 4 different time points. The results were analyzed using SPSS 16.0 software (IBM Corporation, Armonk, NY, USA) with *p* < 0.05.

## 3. Results and Discussion

### 3.1. Electrospinning Using SIS and PCLA

To prepare an SIS solution for electrospinning, the superfine SIS powder was solubilized in the electrospinning solvent HFIP (see the Materials and Methods section). The SIS solution appeared to be slightly suspended. The SIS solution was then subjected to electrospinning to prepare electrospun SIS-NS. The electrospun SIS-NS had irregular and clumping arrangements and a random structure, although the SIS-NS showed overall interconnected fibrillary structures (data not shown). Additionally, the electrospun SIS-NS are easily solubilized in biological media. This result indicated that electrospun SIS-NS could not be used as a drug delivery carrier under biological conditions.

The electrospinning of PCLA solution in HFIP proceeded well, and the electrospun PCLA nanofiber sheets (P-NS) had a well-organized structure. Therefore, PCLA was selected as an interpenetrated, reinforced material to process and maintain the interconnected fibrillary structures of SIS during and after electrospinning. SIS/PCLA (S/P) solution was prepared by adding PCLA to SIS in three ratios. Electrospinning proceeded easily with the addition of PCLA, and the electrospun S/P-NS maintained an appropriate shape (Figure 2a). The experiment confirmed that as the amount of PCLA added to the SIS increased, electrospinning proceeded well. All formulations yielded electrospun S/P-NS with an adjustable thickness and area.

### 3.2. Characterization of Electrospun S/P-NS

Based on the SEM images (Figure 2b), the electrospun P-NS showed interconnected fibrillary structures. All of the electrospun S/P-NS also exhibited a surface with interconnected fibrillary structures, thus indicating a uniform mixture of SIS and PCLA inside the fibrillary structures of the electrospun S/P-NS.

The contact angles of a native SIS sheet, electrospun S/P-NS, and P-NS were measured to analyze the surface properties of the nanofiber sheets (Figure 2c). As described in the previous section, the electrospun SIS was easily soluble in biological media, and thus the contact angles could not be measured. Therefore, a regular SIS sheet was used for the comparison of contact angles with electrospun S/P-NS. The SIS sheet showed contact angles of 40°, indicating a hydrophilic surface property. On the other hand, there was a gradual increase from 69° to 93° in the contact angles of electrospun S/P-NS as the amount of PCLA increased. In addition, electrospun P-NS had contact angles of 115°, indicating a hydrophobic surface property. These results indicated that the ratio of SIS and PCLA could modulate the surface properties of electrospun S/P-NS.

The porosities and pore sizes of electrospun S/P-NS gradually decreased as the PCLA content increased (Figure 2d,e). The electrospun S/P-NS also had increasing diameters ranging from 100 to 360 nm as the PCLA content increased (Figure 2f). Figure 2g shows the ultimate tensile strengths of electrospun S/P-NS. The tensile strengths of electrospun S/P-NS increased as the amount of PCLA increased. The electrospun S/P-NS with a high PCLA content exhibited good physical properties.

The electrospun S/P-NS exhibited a closely compacted network structure with increasing PCLA content. Therefore, with increasing PCLA content, the electrospun S/P-NS had different porosities, pore sizes, and diameters. This was conjectured as the reason that PCLA has better electrospinning performance than SIS. This is consistent with the results that PCLA make electrospinning relatively easy [20,21,22].

Although there was a change in the physical properties of the electrospun S/P-NS based on the ratios of SIS and PCLA, the electrospun S/P-NS was successfully fabricated. Based on the analysis of the morphology, surface property, and tensile strength, an electrospun S/P(5:1)-NS with a high SIS content was selected for application as a drug carrier.

We compared the growth of NIH3 cells on native S-sheet and electrospun S/P-NS. The cells adhered well on native S-sheet and electrospun S/P-NS and well proliferated for 10 days. There was no significant difference in the growth of NIH3 cells among all electrospun S/P-NSs. Therefore, it was confirmed that all electrospun S/P-NS have biocompatibility (Appendix A).

### 3.3. In Vitro Degradation of Electrospun S/P-NS

The in vitro degradation of a native SIS sheet (selected based on the reasons described in the previous section), electrospun S/P-NS, and P-NS was examined by shaking the NS in collagenase and PBS at 37 °C for a predefined period (Figure 3). Degradation of the SIS sheet, electrospun S/P-NS, and P-NS was measured in terms of the weight decrease of the NS as a function of time.

First, the degradation of the SIS sheet, electrospun S/P(5:1)-NS, and P-NS was examined for 3 days. The SIS sheet in the presence of collagenase was completely degraded after 1 day. The electrospun S/P(5:1)-NS showed about 50% degradation at 1 day and remained the same for 3 days. The P-NS showed almost no degradation. This confirmed that the SIS sheet rapidly degraded in the presence of collagenase, but the degradation of P-NS was hardly affected by the collagenase.

The degradation of electrospun S/P(5:1)-NS was examined in different concentrations of collagenase for 3 weeks. The electrospun S/P(5:1)-NS showed rapid degradation at 1 day, and more degradation occurred with increasing amounts of collagenase. After 1 day, the degradation of electrospun S/P(5:1)-NS showed a similar pattern regardless of the amount of collagenase. The degraded products in the presence of collagenase were inferred to be original ingredients of SIS material loaded in the electrospun S/P(5:1)-NS, because PCLA was minimally degraded after 3 weeks.

### 3.4. Electrospinning Using Drug-Loaded S/P(5:1)-NS

Dex and AgS drugs related to anti-inflammation were selected to prepare electrospun Dex-S/P(5:1)-NS and AgS- S/P(5:1)-NS. In the electrospinning procedure, there was little to no structural change in Dex-S/P(5:1)-NS and AgS- S/P(5:1)-NS (Figure 4). In addition, Dex-S/P(5:1)-NS and AgS- S/P(5:1)-NS had an almost uniform shape and fiber diameters (89, 92, 76 nm for no-drug-S/P-NS, Dex-S/P-NS and AgS- S/P-NS, respectively) regardless of drug type.

### 3.5. In Vitro Drug Release from Drug-Loaded S/P(5:1)-NS

To evaluate in vitro Dex and AgS release, Dex-S/P(5:1)-NS and AgS- S/P(5:1)-NS were incubated in PBS at 37 °C for 21 days. Figure 5 shows the Dex and AgS release plots for the cumulative released amounts over time.

The cumulative Dex released in vitro from Dex-S/P(5:1)-NS was approximately 31% at 1 day, 43% at 4 days, 61% at 7 days, and 99% at 14 days. Dex-S/P(5:1)-NSs showed an initial burst of released Dex, likely due to the Dex loaded on the surface of the S/P(5:1)-NS. After an initial burst, the Dex-S/P(5:1)-NS showed a first-order Dex release pattern.

The cumulative AgS released in vitro from AgS- S/P(5:1)-NS also showed an initial burst of AgS at 1 day, although it was a smaller amount compared to the Dex-S/P(5:1)-NS. After the initial burst, AgS- S/P(5:1)-NS showed a gradual AgS release pattern for 6 days. After that, the amount of AgS released was minimal for up to 21 days.

Although Dex-S/P(5:1)-NS and AgS-S/P(5:1)-NS showed different release profiles of Dex and AgS, the drug release was sustained over extended experimental periods. However, Dex-S/P(5:1)-NS exhibited a fast release over a short time compared to the AgS- S/P(5:1)-NS. Kely et al. reported that drug release depends on the relationship such as intrinsic properties of drug and/or polymer, and drug–polymer interaction [27]. Thus it was conjectured that since Dex has a high solubility in water compared to AgS, a higher Dex content led to the faster absorption of water into the S/P(5:1)-NS and thus a more rapid release [28,29].

### 3.6. In Vivo Drug Release from Drug-Loaded S/P(5:1)-NS

The next aim of this study was to analyze Dex-S/P(5:1)-NS and AgS-S/P(5:1)-NS as in vivo, sustained Dex and AgS delivery systems. Dex-S/P(5:1)-NS and AgS-S/P(5:1)-NS were implanted into SD rats and left for 3 weeks. The cumulative amounts of drugs released in vivo were plotted at 3, 7, 14, and 21 days (Figure 6).

At 3 days, 40% of Dex was released from Dex-S/P(5:1)-NS. A linear Dex release pattern was observed over 14 days, and Dex was completely released at 14 days. Importantly, the release pattern over the 14 days was a first-order release profile for in vivo conditions. This was a similar release pattern as the in vitro release of Dex from Dex-S/P(5:1)-NS.

On the other hand, the cumulative amount of AgS released from AgS-S/P(5:1)-NS was 37% at 3 days, and about 3% AgS was released over 21 days, indicating little in vivo release of AgS. This was a similar release pattern as the in vitro release of AgS from AgS-S/P(5:1)-NS.

Overall, the experiment successfully confirmed the possibility for the sustained release of Dex and AgS from the Dex-S/P(5:1)-NS and AgS-S/P(5:1)-NS, respectively, over 3 weeks. In addition, different drug release profiles were observed depending on the drug properties.

### 3.7. Histological Analysis of In Vivo Drug-Loaded S/P(5:1)-NS Implants

The final aim of this study was to evaluate the in vivo compatibility of drug-loaded S/P(5:1)-NS (Figure 7). The implanted drug-loaded S/P(5:1)-NS were individually removed at 1, 2, and 3 weeks. The drug-loaded S/P(5:1)-NS maintained their shape over the entire 3-week experimental period, but their volume gradually decreased as the implantation time increased. Meanwhile, thin fibrous capsules containing fibroblasts and blood vessels were observed around the implanted drug-loaded S/P(5:1)-NS.

Next, the implants were evaluated by H&E and ED1 immunofluorescent staining. H&E staining allowed the implants to be clearly distinguished from the host tissue. The implants showed several blood vessels, and there were negligible differences between the two types of drug-loaded S/P(5:1)-NS.

ED1 immunofluorescent staining revealed red fluorescence and blue fluorescence attributable to the macrophage marker CD68 and the nuclei of live cells, respectively. The red and blue fluorescent signals were observed in all implants.

ED1-positive cells were counted and normalized to the total stained tissue area to determine the extent of macrophage infiltration (Figure 7d). The percentage for Dex-S/P(5:1)-NS decreased to 11% at 1 week, 8% at 2 weeks, and 7% at 3 weeks. The AgS-S/P(5:1)-NS showed 17% macrophage infiltration at 1 week, 8% at 2 weeks, and 12% at 3 weeks. The macrophage infiltration in the no-drug S/P(5:1)-NS-implanted group was 22% at 1 week, 19% at 2 weeks, and 20% at 3 weeks, indicating that there was active infiltration by macrophages.

It was concluded that the Dex-S/P(5:1)-NS and AgS-S/P(5:1)-NS suppressed macrophage infiltration compared to no-drug S/P(5:1)-NS. The Dex-S/P(5:1)-NS showed a gradual decrease in macrophage infiltration, but AgS-S/P(5:1)-NS suppressed macrophage infiltration at 2 weeks and then slightly increased at 3 weeks. This result indicates that sustained drug release is required to suppress macrophage infiltration because Dex and AgS display potent inflammatory suppressive properties [30,31,32].

Overall, these results show that Dex-S/P(5:1)-NS and AgS-S/P(5:1)-NS suppressed macrophage infiltration.

## 4. Conclusions

In this study, electrospun Dex-S/P-NS and AgS-S/P-NS with adjustable thickness and areas were successfully fabricated. Moreover, it was concluded that the Dex-S/P-NS and AgS-S/P-NS successfully sustain the in vivo release of Dex and AgS over a defined implantation period and thus suppress macrophage infiltration. Further experiments are now in progress to investigate the feasibility of in vivo applications for specific diseases in animal models.

## Figures and Tables

**Figure 1 pharmaceutics-13-00253-f001:**
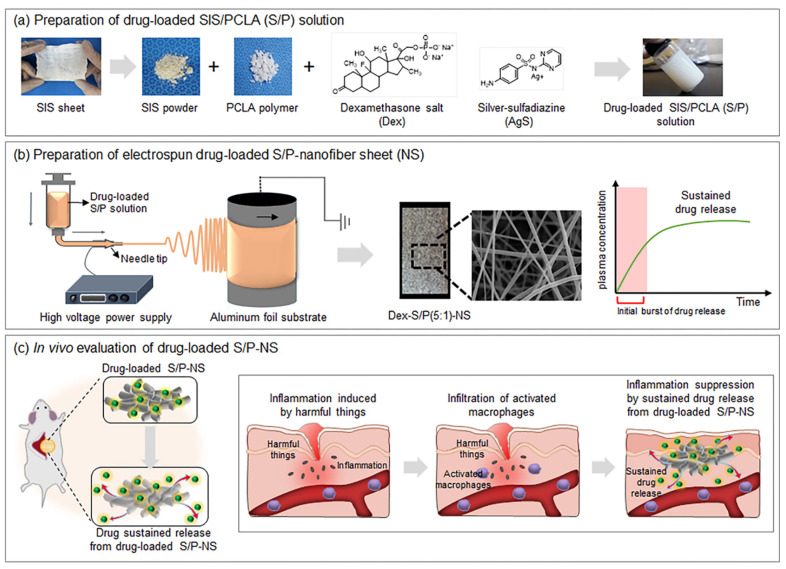
Schematics of the experimental procedures for (**a**) preparation of drug-loaded small intestine submucosa (SIS)/ poly(ε-caprolactone-*ran*-l-lactide) (PCLA) (S/P) solution, (**b**) preparation of electrospun drug-loaded SP nanofiber sheet (NS), and (**c**) in vivo evaluation of drug-loaded SP-NS.

**Figure 2 pharmaceutics-13-00253-f002:**
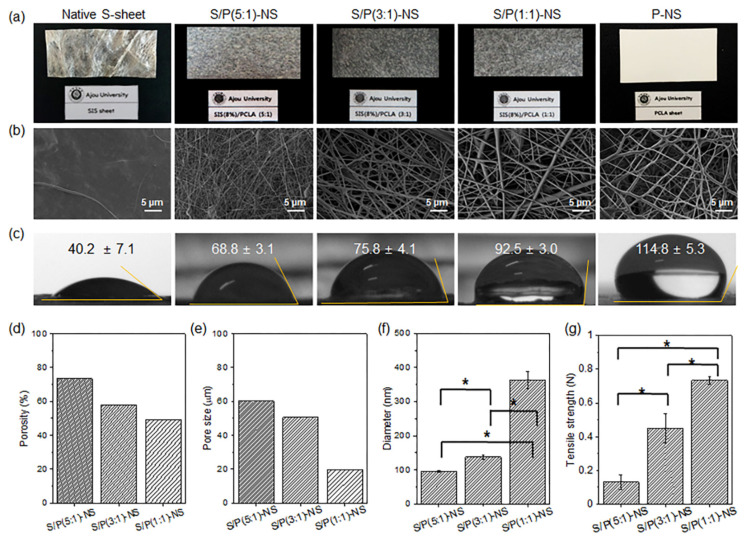
(**a**) Optical images, (**b**) SEM images (3000× magnification, 5 µm), (**c**) contact angles of native SIS sheet, S/P(5:1)-NS, S/P(3:1)-NS, S/P(1:1)-NS, and P-NS, and (**d**) porosities, (**e**) pore sizes, (**f**) diameters, and (**g**) tensile strengths of electrospun S/P(5:1)-NS, S/P(3:1)-NS, and S/P(1:1)-NS (* *p* < 0.05).

**Figure 3 pharmaceutics-13-00253-f003:**
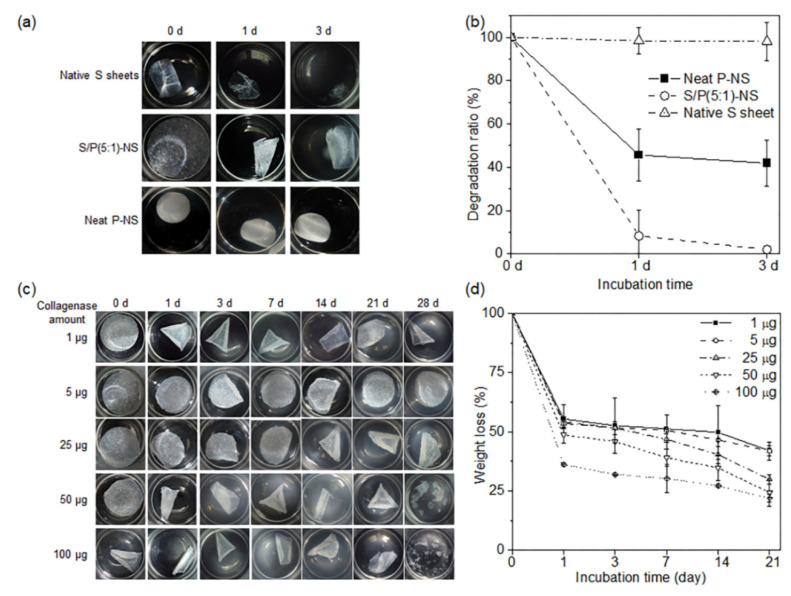
Biodegradation of S/P(5:1)-NS in the presence of collagenase: (**a**) Optical images and (**b**) plotted degradation ratios of native SIS (S)-sheet, S/P(5:1)-NS, and P-NS in the presence of 50 µg collagenase for 3 days, (**c**) optical images, and (**d**) plot of degradation ratios of S/P(5:1)-NS in the presence of different collagenase quantities over 21 days.

**Figure 4 pharmaceutics-13-00253-f004:**
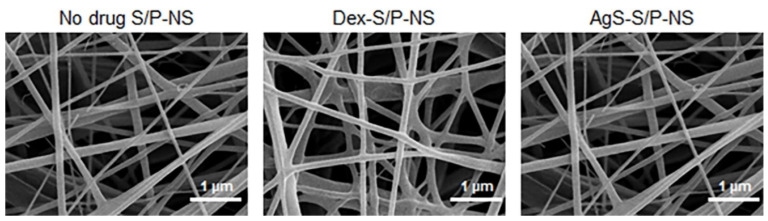
SEM images (3000 × magnification, 1 µm) of electrospun no-drug-S/P(5:1)-NS, Dex-S/P(5:1)-NS, and AgS-S/P(5:1)-NS.

**Figure 5 pharmaceutics-13-00253-f005:**
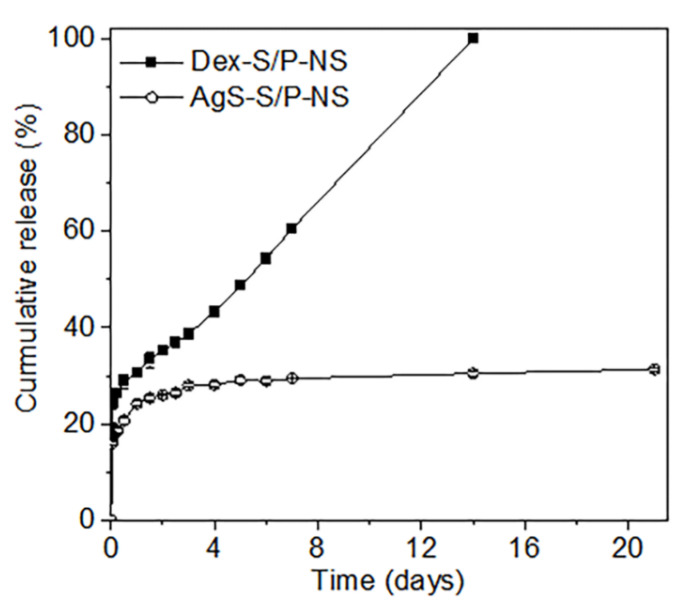
In vitro cumulative release profiles of Dex from Dex-S/P-NS and AgS from AgS-S/P-NS over 21 days.

**Figure 6 pharmaceutics-13-00253-f006:**
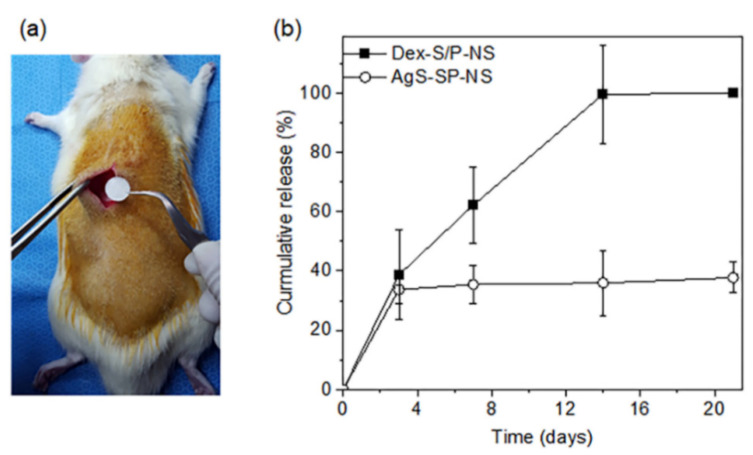
(**a**) Animal implantation and (**b**) in vivo cumulative release profiles of Dex from Dex-S/P-NS and AgS from AgS-S/P-NS over 21 days.

**Figure 7 pharmaceutics-13-00253-f007:**
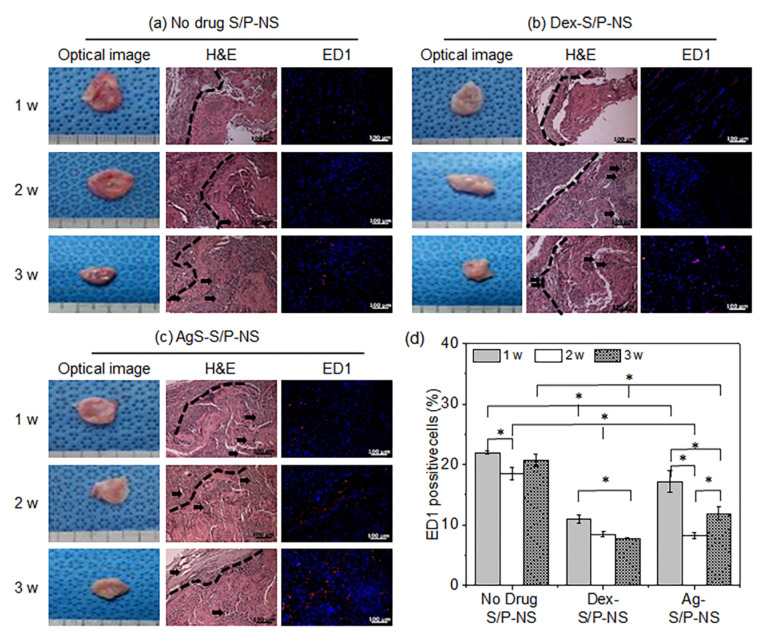
In vivo evaluation of S/P-NS without and with drug: Optical images, H&E (200× magnification, 100 µm), and ED1 (200× magnification, 100 µm) staining images of (**a**) no-drug SP-NS, (**b**) Dex-S/P-NS, (**c**) AgS-S/P-NS, and (**d**) ED1 positive ratios for each NS at 1, 2, and 3 weeks (* *p* < 0.05) (in H&E, arrows indicated blood vessels).

## Data Availability

The supporting data presented in this study are available in the Supplementary Material.

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
