# Peer review of "Preparation of Electrospun Small Intestinal Submucosa/Poly(caprolactone-co-Lactide-co-glycolide) Nanofiber Sheet as a Potential Drug Carrier"

_pharmaceutics, 2021, doi:10.3390/pharmaceutics13020253_

Round 1

Reviewer 1 Report

Dear Editor, in the present work drug-loaded electrospun intestine submucosa (SIS) nanofiber sheet have been prepared. A polymer matrix poly(ε-caprolactone-ran-L-lactide) (PCLA) copolymers has been used, to improve the electrospinning performance of SIS. The electrospun nanofibers were used as drug carrier of dexamethasone (Dex) and silver sulfadiazine (AgS) drug related to anti-inflammation. The paper is well organized and provides new data. For this reason, I propose to accept it for publication. In following you can find some minor importance comments.

What is the physical state of drug in nanofibers? XRD or DSC data should be provided.

What is the dispersion of AgS in nanofibers?

AgS was not completely released. Is this due to the release of AgS attached in nanofibers surface?

Author Response

Responses to the comments by Reviewer 1

We appreciate the reviewer’s comments. We have addressed each of these comments below and have highlighted revisions made to the relevant passages of the manuscript.

1) What is the physical state of drug in nanofibers? XRD or DSC data should be provided.

We appreciate the reviewer’s comment. We did not measure XRD and DSC. We believed that the drug in the nanofibers maintained a stable structure. The reason was that drug were detected in the similar position as the initial peak in HPLC of the drug after release experiment and additionally, it was also confirmed that although the released amount was very small after release experiment, drug maintained the similar NMR structure of original drugs.

2) What is the dispersion of AgS in nanofibers?

We appreciate the reviewer’s comment. If the reviewer’s comment is “What form or how the AgS is dispersed in the nanofibers”, we believed that the AgS was distributed in the nanofiber throughout. The reason is that since electrospinning was performed by using a solution of AgS and SIS/PCLA mixture, the AgS can be distributed in the nanofibers throughout.

3) AgS was not completely released. Is this due to the release of AgS attached in nanofibers surface?

We appreciate the reviewer’s comment. In the current study, it was difficult to accurately determine the reason for low release. However, we believed that low AgS release was due to the low solubility of AgS and the formation of hydrophobic interaction between AgS and SIS/PCLA.

We again appreciate the reviewer’s useful suggestions and comments. We have revised the manuscript to be in line with the reviewer’s comments as much as possible. We would again appreciate your kind consideration to our paper’s publication in Pharmaceutics.

Reviewer 2 Report

In general, this manuscript is well written, with scientific soundness, very interesting and original. The existing methods are well described, but some questions need to be addressed before publication 

Comments and suggestions

  1. Please add the examples of the biomedical products mentioned in the introduction section (Page 1, lines 33, 34)
  2. Before in vivo testing, you should test your samples with an in vitro cellular cytotoxicity test even if the raw materials are approved by the FDA, did you perform these types of tests in previous work?
  3. Please revise the whole document some in vivo and in vitro expressions are not in italics
  4. Did you measure the average fiber diameter of the drug-loaded S/P-NS?.
  5. Did you physiochemically characterize your fibers in previous work (FTIR, TGA-DSC, HNMR, HPLC), how you proved you have the SIS incorporated in the PCLA fibers, with the water contact test?
  6. In my point of view, your results are well described, but it needs more discussion, your results need to be compared and your comments need to be validated with the reported in the literature, please add more references in the results and discussion section
  7. Please revise the whole document, some typing errors were found
  8. In the histological images, you should add the controls to see the differences between the normal tissue and the samples exposed tissues in figure 7

Author Response

Responses to the comments by Reviewer 2

We appreciate the reviewer’s comments. We have addressed each of these comments below and have highlighted revisions made to the relevant passages of the manuscript.

1) Please add the examples of the biomedical products mentioned in the introduction section (Page 1, lines 33, 34)

We added the biomedical products in line 33-34.   

2) Before in vivo testing, you should test your samples with an in vitro cellular cytotoxicity test even if the raw materials are approved by the FDA, did you perform these types of tests in previous work?

We think that the reviewer’s comment is well founded. We have already examined the cytotoxicity test using nanofiber and thus added in Figure S2 as supporting information. 

3) Please revise the whole document some in vivo and in vitro expressions are not in italics

We appreciate the reviewer’s comment. We revised thorough in the manuscript.

4) Did you measure the average fiber diameter of the drug-loaded S/P-NS?

The reviewer’s point is well taken. We have already measured and added in the lines 323-324.

5) Did you physiochemically characterize your fibers in previous work (FTIR, TGA-DSC, HNMR, HPLC), how you proved you have the SIS incorporated in the PCLA fibers, with the water contact test?

We confirmed that ratios of the content of the insoluble portion and soluble portion in CH2Cl2 was almost 5 versus 1 as below. The insoluble portion was assumed to be SIS and the soluble portion in CH2Cl2 was confirmed as PCLA in NMR (Figure S2). So, we added the explanation in experimental sections lines 121-124. Following tables are the raw data.

Table. Raw data in content of the insoluble portion and soluble portion in CH2Cl2

Sheet

Weight

Theoretical

ratio

Average

Dried weight

Average after dry

SIS

62.6

52.17

49.28

38.8

36.8

57.4

47.83

36.1

57.4

49.25

35.5

PCLA

62.6

10.43

9.86

8.6

8.07

57.4

9.57

8.5

57.4

9.85

7.1

Table. Ratios of SIS and PCLA in the electrospun S/P(5:1)-NS

No.

SIS weight

PCLA weight

SIS:PCLA (w:w)

1

38.8

8.6

4.51:1

2

36.1

8.5

4.25:1

3

35.5

7.1

5:1

6) In my point of view, your results are well described, but it needs more discussion, your results need to be compared and your comments need to be validated with the reported in the literature, please add more references in the results and discussion section

The reviewer’s point is well taken. We added the discussion of the results compared with the other groups and added additional references.

7) Please revise the whole document, some typing errors were found

We appreciate the reviewer’s comment. We revised thorough in the manuscript.

8) In the histological images, you should add the controls to see the differences between the normal tissue and the samples exposed tissues in figure 7

We appreciate the reviewer’s comment. In current experiment, we used no-drug S/P-NS as controls to see the differences between the normal tissue and the samples exposed tissues. We will use the control in the next experiment.

We again appreciate the reviewer’s useful suggestions and comments. We have revised the manuscript to be in line with the reviewer’s comments as much as possible. We would again appreciate your kind consideration to our paper’s publication in Pharmaceutics.

Round 2

Reviewer 2 Report

The authors had successfully resolved all my doubts and concerns. I recommend this article for publication in Pharmaceutics

Recognizing this manuscript have enough quality to be published in the present form

Author Response

Responses to the comments by Reviewer 1

We appreciate the reviewer’s comments. We have addressed each of these comments below and have highlighted revisions made to the relevant passages of the manuscript.

We appreciate the reviewer’s comment. We did not measure DSC in the original experiment.

According to the reviewer’s useful suggestions and comment, we measured H-NMR, DSC and HPLC for the relevant experiments and then added the corresponding results in the Figures S2-6.

We again appreciate the reviewer’s useful suggestions and comments. We have added the relevant Figure in the supporting information. We would again appreciate your kind consideration to our paper’s publication in Pharmaceutics.
